# Diffusion Behavior of Chloride Ions in Concrete Box Girder under the Influence of Load and Carbonation

**DOI:** 10.3390/ma13092117

**Published:** 2020-05-02

**Authors:** Yinglong Liu, Pengzhen Lin, Junjun Ma

**Affiliations:** Laboratory of Road & Bridge and Underground Engineering of Gansu Province, Lanzhou Jiaotong University, Lanzhou 730070, China; liuyinglong23@163.com (Y.L.); majjlz@163.com (J.M.)

**Keywords:** concrete box girder, carbonation, chloride corrosion, durability characteristics

## Abstract

In order to study the durability degradation characteristics of concrete box girder under load and carbonation and chloride ion erosion, a scale model of concrete box girder was made for experimental research. According to the test results, the diffusion characteristics of chloride ions in the concrete box girder under the coupling effect of load and carbon dioxide were analyzed. By revising the calculation formula of the existing chloride ion concentration considering multiple factors, a calculation model of chloride ion concentration considering the influence of carbonation was proposed, and the test results were verified. The results show that the chloride concentration of the box girder on the same cross section is non-uniformly distributed due to the shear lag effect and the spatial structure. After considering the effect of carbonation, the difference rate of the improved model proposed in this paper is generally within 10%. Compared with the original model, the difference rate is reduced by a maximum of 19%.

## 1. Introduction

The durability of concrete structures involves structural safety, which has become a key issue in the engineering community [1]. The most prominent problem affecting the durability of concrete structures is the corrosion of steel bars [2], and concrete carbonation and chloride ion erosion are the main causes [2,3]. In northern areas, CO_2_ and Cl^−^ erosion are the most common and coupling corrosion types of concrete bridges [3]. Once the steel bars in the concrete corrode, not only will the effective area of the steel bars be reduced, but the resulting corrosion products will also expand the volume of the steel bars, which will cause the concrete protective layer to crack or even fall off, which will eventually lead to a decrease in the durability and bearing capacity of the concrete structure [4,5]. Studies in the literature of Zhu X. J. et al. [6] have shown that the life of concrete structures under the combined action of carbonation and chloride ions has been reduced by nearly 20 years. In 2004, Germany spent about 300 million euros on highway maintenance and repairs, of which 40% was used to repair concrete bridge damage caused by chloride salt corrosion and steel bar corrosion [7]. The annual cost of corrosion in China since 2014 has been approximately 3.34% of GDP [8]. Northern Europe, Arabian Gulf and other regions are also facing serious steel corrosion problems [9]. With the increase in the frequency of structural accidents due to insufficient durability, the durability design of concrete structures considering carbonation and chloride ion corrosion has gradually been reflected in the relevant design codes of various countries [10,11,12,13].

Current research on the durability of concrete structures is mainly based on solid beams (rectangular, trapezoidal, etc.) [3,5]. The difference to the solid section beam is that the box beam is a hollow structure composed of a relatively small thickness of the top plate, web plate and bottom plate. This makes the characteristics of box girder eroded by carbonation and chloride ion have geometric and mechanical spatial multi-dimensionality. The specific manifestation is that the outer wall and inner cavity of the box girder have non-uniform temperature, humidity and air (wind) pressure distribution. In addition, the box girder is mainly subjected to bending under the action of moving live loads (cars, trains) and its own weight. Due to the thin-walled characteristics and the shear lag effect [14,15,16], the stress of the same section of the box girder is unevenly distributed. This will inevitably cause the carbonation and chloride ion erosion of the box girder structure to be a spatially non-uniform degradation mode. Researches on concrete carbonation and chloride ion corrosion have obtained the carbonation and chloride ion corrosion mechanisms, laws and service life prediction methods that consider different material compositions, different environmental conditions, different loading conditions, and coupling effects [17,18,19,20,21,22,23,24,25,26,27,28,29]. A method for reducing the corrosion of concrete by carbonation and chloride ions in terms of material composition, protective layer thickness, concrete strength, chloride ion permeability coefficient, structural system, and construction measures has been initially formed. However, existing research is mainly aimed at concrete solid members. For the most common concrete box girder in highways, railways and municipal bridges, considering the specific structure of the box girder and the differences between the indoor and outdoor environments, it is very important to establish the mechanism and prediction method of the concrete box girder corroded by carbonation and chloride ions. Existing studies have shown that the diffusion of carbon dioxide and chloride ions in concrete has a strong correlation. The effect of carbonation on the chloride ion’s effect on the erosion rate of concrete has two aspects: inhibition and promotion. Nagla and Page [30] carbonation causes the porosity inside the concrete to decrease. As a result, the diffusion rate of chloride ions decreases. However, carbonation will change the pore size distribution of the concrete and increase the pores [31]. In addition, carbonation will release free water, causing local pore saturation to change [32]. This will further accelerate the diffusion of chloride ions [33]. On the other hand, the carbonation reaction will reduce the adsorption capacity of concrete to chloride ion [34]. Therefore, carbonation causes a part of the bound chloride ion to be absorbed by the concrete to be released, changing the local chloride ion concentration gradient [35] and accelerating the diffusion of chloride ions [36]. The main effect of carbonation on the diffusion of chloride ions is to reduce the binding capacity of concrete to chloride ions [37]. The results of literature [32] also show that carbonization will accelerate the corrosion of chloride ions to concrete, ignoring the impact of carbonation on chloride ions, which will lead to unsafe durability design.

Therefore, based on the effect of carbonation on the binding capacity of concrete to chloride ions, this paper derives a calculation model of chloride erosion considering the effect of carbonation. By considering the load and the spatial structural characteristics of the box girder, the formula was modified to obtain a chloride ion calculation formula suitable for the box girder structure considering the effect of carbonation load and verified with the test results. This paper also studies the characteristics of chloride ion erosion of the concrete box girder structure under the action of carbonation and load.

## 2. Methods

### 2.1. Calculation Formula of Chloride Concentration Correction Considering Carbonation

Yu H. F. [38] combined the research results of Thomas M D A [17], Prezzi M. [18] and Mangat P S, et al. Mangat, P.S. [19] and Mangat, P. [20] proposed the chloride ion diffusion equation of concrete under the action of multiple factors:
(1)$$C_{f}=C_{0}+(C_{s}-C_{0})\left[1-erf\frac{x}{2\sqrt{\frac{KD_{0}^{\prime }Tt_{0}^{m}}{(1+R)(1-m)T_{0}}e^{q(\frac{1}{T_{0}}-\frac{1}{T})}t^{1-m}}}\right]$$
where *K* is the degradation effect coefficient of the chloride ion diffusion performance of the concrete, $D_{0}^{\prime }$ is the chloride ion diffusion coefficient of concrete measured at the hydration age of *t*_0_, *m* is an empirical constant , and *q* is the activation constant related to the water-cement ratio. Stephen et al. [24] explained the value of *q* at different water–cement ratios. At time *t*, the chloride ion concentration is *C_t_*, the free chloride ion concentration is *C_f_*, and the combined chloride ion concentration is *C_b_*, then the chloride ion binding capacity *R* [21,26,29] of the concrete can be expressed as:(2)$$R=\frac{C_{b}}{C_{f}}=\frac{C_{t}-C_{f}}{C_{f}}$$

R includes three aspects [31]: chloride ions are chemically combined into C_3_A. CaCl_2_∙12H_2_O by C_3_AH_5_, enter the CSH gel structure and are physically adsorbed on the internal pore surface of concrete.

Affected by the concrete’s ability to bind to chloride ion *R*, under the combined action of CO_2_ and Cl^−^, CO_2_ can cause the following chemical reactions of the combined chloride salts [32]. The interaction between chloride ion and carbon dioxide in concrete is shown in Figure 1. 3CaO∙Al_2_O_3_∙CaCl_2_∙10H_2_O + 3CO_2_→3CaCO_3_ + Al_2_O_3_ + Ca^2+^ + 2Cl^−^ + 10H_2_O.

It can be seen that carbonation can destroy the combined hydrated chloric acid, so that the free chloride ions that are bound by the chloride salt during the concrete diffusion process are released again. Therefore, carbonation mainly affects the binding ability of concrete to chloride ions. In order to quantify the effect of carbonation on the diffusion of chloride ions, carbonation influence coefficient *λ* was introduced to reflect the effect of carbonation on the ability of concrete to bind chloride salt *R*.

According to Equation (1), after considering the carbonation influence coefficient *λ*, the free Cl^−^ concentration at the carbonation front of the concrete can be expressed as:(3)$$C_{f}=C_{0}+(C_{s}-C_{0})\left[1-erf\frac{x}{2\sqrt{\frac{KD_{0}^{\prime }Tt_{0}^{m}}{(1+\lambda R)(1-m)T_{0}}e^{q(\frac{1}{T_{0}}-\frac{1}{T})}t^{1-m}}}\right]$$

Without considering the binding ability *R* of concrete to chloride ion, let *R* = 0, and the concentration of chloride ion is:(4)$$C_{f}=C_{0}+(C_{s}-C_{0})\left[1-f(x)\right]$$
where f(x) is the error function, $f(x)=erf\left(\frac{x}{2\sqrt{\frac{KD_{0}^{\prime }Tt_{0}^{m}}{(1-m)T_{0}}e^{q(\frac{1}{T_{0}}-\frac{1}{T})}t^{1-m}}}\right)$.

The Equation (4) is subtracted from the Equation (3) to obtain the free chloride ion concentration *C_C_* released from the carbonized concrete:(5)$$C_{c}=(C_{s}-C_{0})\left[f_{\lambda }(x)-f(x)\right]$$
where $f_{\lambda }(x)$ is the error function considering the carbonation influence coefficient, $f_{\lambda }(x)=erf\left(\frac{x}{2\sqrt{\frac{KD_{0}^{\prime }Tt_{0}^{m}}{(1+\lambda R)(1-m)T_{0}}e^{q(\frac{1}{T_{0}}-\frac{1}{T})}t^{1-m}}}\right)$.

On the basis of Equation (1), the free chloride ion concentration *C_C_* released by the concrete in the carbonation zone is superimposed, and the calculation formula of the free chloride ion concentration at the front of the carbonation of the concrete after considering the impact of carbonation can be obtained as:(6)$$C_{f}=C_{0}+(C_{s}-C_{0})\left[1-f_{R}(x)+f_{\lambda }(D)-f(x)\right]$$
where *D* is the carbonation depth, *λ* is the carbonation influence coefficient, and the value ranges from 0 to 1. $f_{R}(x)$ is the error function considering the binding capacity of concrete to chloride ions, $f_{R}(x)=erf\left(\frac{x}{2\sqrt{\frac{KD_{0}^{\prime }Tt_{0}^{m}}{(1+R)(1-m)T_{0}}e^{q(\frac{1}{T_{0}}-\frac{1}{T})}t^{1-m}}}\right)$.

At present, when a large amount of literature studies the effect of external load on the chloride ion diffusion effect in concrete, the effect of load on the chloride ion diffusion effect is generally considered by multiplying the original chloride ion diffusion coefficient by the corresponding influence coefficient [39,40]. Under the bending load, the relationship between the stress distribution of the concrete section and the chloride diffusion coefficient in the concrete is shown in Equation (7) [33,34]:(7)$$D_{\sigma }=f\left(\eta \right)\cdot D_{0}$$
where $D_{\sigma }$ is the Cl^−^ diffusion coefficient in the concrete after considering the load, f(η) is the influence coefficient of the load on the chloride ion diffusion coefficient, and *D*_0_ is the chloride ion diffusion coefficient in the concrete under no stress.

In order to study the effect of external loads on the diffusion of chloride ions, the specific effects of tensile and compressive stresses were obtained in literatures [39,40] through regression analysis of experimental data.

The expression under tensile stress is:
(8)$$f\left(\eta \right)=1.0+0.9598\eta_{t}-0.3608\eta_{t}^{2}$$

The expression under compressive stress is:(9)$$f\left(\eta \right)=1.0-1.6626\eta_{c}+2.2560\eta_{c}^{2}$$

Combining the common Equations (8) and (9) to modify the chloride ion diffusion coefficient in the Equation (6), the analysis of the chloride ion diffusion effect of the concrete structure under load and carbonation can be realized at the same time.

Therefore, the formula for calculating the chloride ion concentration of concrete under carbonation and load is:(10)$$C_{f}=C_{0}+(C_{s}-C_{0})\left[1-f_{R\sigma }(x)+f_{\lambda }{}_{\sigma }(D)-f_{\sigma }(x)\right]$$
where $f_{R\sigma }(x)$ is an error function that considers the effect of stress and the ability of concrete to bind to chloride ions, $f_{\lambda \sigma }(D)$ is an error function that considers the effect of stress and carbonation, and $f_{\sigma }(x)$ is the error function considering the effect of stress. $f_{R\sigma }(x)=erf\left(\frac{x}{2\sqrt{\frac{KD_{0}^{\prime }Tt_{0}^{m}}{(1+R)(1-m)T_{0}}e^{q(\frac{1}{T_{0}}-\frac{1}{T})}t^{1-m}}}\right)$; $f_{\lambda \sigma }(D)=erf\left(\frac{D}{2\sqrt{\frac{KD_{0}^{\prime }Tt_{0}^{m}}{(1+\lambda R)(1-m)T_{0}}e^{q(\frac{1}{T_{0}}-\frac{1}{T})}t^{1-m}}}\right)$; $f_{\sigma }(x)=erf\left(\frac{x}{2\sqrt{\frac{KD_{0}^{\prime }Tt_{0}^{m}}{(1-m)T_{0}}e^{q(\frac{1}{T_{0}}-\frac{1}{T})}t^{1-m}}}\right)$.

The box girder has a hollow section, and there are multiple two-dimensional diffusion regions in the same section. According to Fick’s second law, the two-dimensional diffusion equation of chloride ion is:(11)$$\frac{\partial^{2}C}{\partial x^{2}}+\frac{\partial^{2}C}{\partial y^{2}}=\frac{1}{D}\frac{\partial C}{\partial t}$$

Assuming the boundary conditions and initial conditions are:

$C|{}_{t>0,x=0,y=0}=C_{s},\;C|{}_{t=0,x>0,y>0}=C_{0}$, by applying Laplace transform to Equation (11), the analytical solution of the two-dimensional chloride diffusion equation can be obtained as:(12)$$C_{f}=C_{0}+\left(Cs-C_{0}\right)\left(1-erf\left(\frac{x}{2\sqrt{Dt}}\right)erf\left(\frac{y}{2\sqrt{Dt}}\right)\right)$$

According to Equation (12), after considering the two-dimensional diffusion, the chloride ion concentration calculation formula can be expressed as:(13)$$C_{f}=C_{0}+(C_{s}-C_{0})\left[1-f_{R\sigma }(x)f_{R\sigma }(y)+f_{\lambda }{}_{\sigma }(D_{x})f_{\lambda }{}_{\sigma }(D_{y})-f_{\sigma }(x)f_{\sigma }(y)\right]$$
where $D_{x}$ and $D_{y}$ are the carbonation depths in the x and y directions, respectively.

### 2.2. Chloride Ion Erosion Test of Single Box and Single Chamber Box Girder Considering the Effect of Carbonation and Load

#### 2.2.1. Test Scheme

Deterioration tests of concrete box girder considering load and carbonation and chloride ion erosion were carried out in three groups. The load, carbonation and chloride ion corrosion were considered respectively. The specific test scheme is shown in Table 1.

#### 2.2.2. Model Size and Measurement Point Arrangement

The concrete carbonation test box girder model is a single-box single-chamber equal-section box girder model with a length of L = 60 cm. The roof width and plate thickness were 45 cm and 3 cm, respectively.

For the measurement of chloride ion concentration after carbonation and chloride ion erosion, five measuring points are set on the top plate of the box beam span and three measuring points are set on the bottom plate. One measuring point is set on the neutral axis position of the webs on both sides.

When loading a concrete box girder, the magnitude of the applied stress is measured by a strain gauge attached to the surface of the concrete box girder model.

The measuring points of the mid-section of the box girder and the arrangement of the strain gauges are shown in Figure 2.

#### 2.2.3. Concrete Proportioning Design and Model Making

The concrete strength of this experimental box girder is C50. The performance of the selected 52.5 ordinary Portland cement products is shown in Table 2.

The test sand was natural river sand with a mud content and fineness modulus of 2.15. Bulk density and apparent density are 2623 kg/m^3^. The coarse aggregate is crushed stone with a particle size ranging from 5 mm to 15 mm. The crushing index and bulk density are 7.0% and 1493 kg/m, respectively. Water is laboratory tap water. The mixing ratio is shown in Table 3.

The performance indexes of FDN water reducer are shown in Table 4.

A forced mixer is used to mix the concrete. Weigh sand, cement, crushed stone according to the corresponding mix ratio accurately, then put it into the mixer for dry mixing for 2 min. After completing the above operations, pour in the weighed water and water reducer and stir for 2 min to make a concrete mixture. To ensure the compactness of the box girder model, a small vibrating rod is used to vibrate when pouring concrete.

After the model is prepared, geotextile is used to ensure the humidity and temperature of the maintenance. After the model is left for 10 h, the formwork is removed. Subsequently, the concrete box girder model is placed in a standard curing room for 28 days. The test piece manufacturing process is shown in Figure 3.

## 3. Results and Discussion

The content of free chloride ion is tested by chemical chloride testing. The specific test method can be referred to in [41]. After the chloride ion erosion, a 12 mm drill bit is used for borehole sampling, each sample is not less than 6 grams, and 20 samples are taken for each test point; the average value is taken as the free chloride ion concentration of the test point. In this test, samples are taken at two depths of 4 mm and 7 mm from the surface. The carbonized corroded box beam model is cut along the mid-span section and sprayed with phenolphthalein to test the depth of carbonation after the extraction of concrete samples containing chloride ions at all test points.

According to China’s national industry standard Code for Design of Concrete Structures [42], the ft of C50 concrete is taken to be 1.89 MPa. A torque wrench is used to tighten the bolts according to the fixed torque to achieve the pulling of the two box beam models. During loading, a dial indicator is used to test the deflection of the top and bottom plates, and the deflection measuring points of the top and bottom plates are arranged at No. 2, No. 4, No. 6 and No. 8 respectively. The strain gauges arranged at the measuring points 6–8 of the bottom plate are mainly used to test the magnitude of the applied tensile stress. Due to the test error, it is difficult to ensure that the strains of the measuring points 6–8 reach 1.5 MPa at the same time. In order to avoid the cracking of the bottom plate due to excessive tensile stress, the loading is stopped immediately when the tensile stress at any of the measurement points (No. 6–8) reaches 1.5 MPa. The compressive strength of the three concrete cube test blocks of standard curing (temperature and humidity are 20 °C ± 2 °C and 95% respectively). after 28 days and the actual vertical deformation and stress test results of each measuring point after loading are shown in Table 5.

After the concrete box girder model reaches the time of CO_2_ and Cl^−^ corrosion, the free Cl^−^ concentration test results at 4 mm and 7 mm from the top, bottom, and web surfaces of the concrete box girder model are shown in Table 6.

### 3.1. Effect of Load and Carbonation on Chloride Diffusion

In order to eliminate the effect of test dispersion on the analysis results as much as possible when analyzing the effect of carbonation and stress on the diffusion of chloride ions, the average value of the chloride ion concentration at each test point of the top plate, bottom plate and web plate were taken as the chloride ion concentrations at the corresponding parts under the same erosion conditions. For example, the erosion condition of 1 # beam and 2 # beam is 0.8ft + CA + CH, so the chloride ion concentration of the top plate under this erosion working condition is the average value of the test points of 1 # beam No. 1–5 and the test points of 2 # beam No. 1–5.

#### 3.1.1. Effect of Load on Chloride Diffusion

In order to analyze the influence of load on the chloride diffusion of the box girder, the average chloride concentrations of 3# beam and 4# beam top, bottom and web under the action of 0.8 ft + CH are compared with that of 5# beam corresponding to CH only. To reflect the uneven degree of chloride ion distribution, the standard deviation (σ)under different exposure conditions is provided. The results are shown in Figure 4.

As shown in Figure 4, at 4 mm and 7 mm respectively, the chloride ion concentration of the top plate under compressive stress is less than that under no load. The chloride ion concentration of the bottom plate under tensile stress is greater than the chloride ion concentration of the bottom plate under no stress. The maximum decrease (increase) of chloride ions under compressive (tensile) stress is about 17%. At 4 mm and 7 mm away from the surface of the box girder, the standard deviation (σ) of the chloride ion concentration under load is large, indicating that the load action increases the uneven degree of chloride ion distribution. In addition, compressive stress can suppress the diffusion of chloride ions, and the tensile stress promotes the diffusion of chloride ions in concrete.

#### 3.1.2. Effect of Carbonation on Chloride Diffusion

In order to analyze the effect of carbonation on the diffusion of chloride ions, the average chloride ion concentrations in the roof, floor and web under 0.8ft + CA + CH erosion conditions were compared with the average chloride ion concentrations in corresponding positions under 0.8ft + CH erosion conditions. The results are shown in Figure 5.

From Figure 5 we can see that at 4 mm and 7 mm away from the surface of the box girder, under the exposure conditions of 0.8ft + CA + CH and 0.8ft + CH, the standard deviations (σ) of the chloride ion concentration at the top, bottom, and web are generally close, and the standard deviation of the chloride ion concentration additionally subjected to carbonation is slightly larger. When the unevenness of the chloride ion concentration of the two exposure conditions in each part of the box girder is close, the chloride ion concentration of the box girder roof and bottom plate corroded by additional carbon dioxide is significantly increased, and the increment is between 23.5% and 33.5%. The increment of chloride ion concentration in the web under carbonation is 16.6% and 26.7%, respectively. Therefore, carbonation greatly promotes the diffusion of chloride ions in the box girder.

#### 3.1.3. Effect of Shear Lag on Chloride Diffusion in Box Girder

The box girder is different from the general solid section due to the space thin-wall effect. The longitudinal stress on the cross section under the bending load is non-uniformly distributed. This phenomenon is the shear lag effect of the box girder. Theoretically, under the load condition of 0.8ft for this test box beam, the theoretical stress distribution of the top and bottom plates is shown in Figure 6.

Under the action of 0.8ft load, the actual distribution of chloride concentration along the top and bottom plate of each measuring point of the model box girder is shown in Figure 7.

The top plate of the model box girder in this paper is subjected to compressive stress under load. It can be seen from Figure 7a,b, the distribution law of the chloride ion concentration at each measuring point on the top plate at 4 mm and 7 mm away from the surface is consistent with the distribution law of the shear lag effect under the corresponding load conditions in Figure 6a. The Cl^−^ concentration at the top of the box girder under the load is subject to "shear lag distribution". Combined with the structural characteristics of the box girder, there is a two-dimensional diffusion of chloride ions at measuring points 6 and 8 of the bottom plate. The results of shear lag effect distribution in Figure 6b under the same load condition show that the maximum tensile stress also occurs at No. 6 and No. 8 measuring points. The distribution law of the actual chloride concentration in Figure 7c,d is also consistent with the shear lag effect of the box girder floor. Therefore, affected by the spatial structural characteristics of the box girder and the shear lag effect, the Cl^−^ concentration of the box girder is unevenly distributed on the same cross section.

### 3.2. Verification of Calculation Model for Chloride Ion Erosion of Concrete Box Girder

In order to eliminate the discreteness of the test results as much as possible, the measured values of the same measuring point of the two box girders under the same working condition are taken as the test values of the corresponding measuring points under the same working condition. For example, the load case of 1#beam and 2# beam is 0.8ft + CA + CH. Under this condition, the test value of the No. 1 measuring point is the average value of the No. 1 measuring point of 1# beam and No. 1 measuring point of 2# beam. Equation (13) is used to calculate the chloride ion concentration at each measuring point of the test box beam under different erosion conditions at 4 mm and 7 mm from the surface. The calculation accuracy of Equation (13) is analyzed by comparing with the measured value. According to the values in [38], $D_{0}^{\prime }$ of 28-Day is 2.5 cm/a. The temperature is 20 °C. Because the model is a complete structure just after 28 days of standard curing, the structural damage coefficient *K* is taken as 1.0. The carbonation influence coefficient *λ* in the complete carbonation area is taken as 0.3. When there is no carbonation, *λ* is taken as 1.0. *R* and *m* are taken as 2.0 and 0.64 respectively.

For the convenience of analysis, the difference rate *η* is introduced, and the expression is:(14)$$\eta =\frac{Formula\;calculated\;value-Test\;value}{Formula\;calculated\;value}\times 100\%$$

In order to make all the differences positive, all the differences calculated by each model are taken as absolute values. The Equation (14) is used to calculate the difference between the measuring points at 4 mm and 7 mm away from the surface of the box beam. The results are shown in Figure 8.

As shown in Figure 8a, 4 mm away from the surface of the box girder, under the action of 0.8ft + CH, the difference rate between the measurement points 3 and 4 is about 14%, and the difference rate between the other measurement points is within 10%. When corroded only by chloride ion, the difference rate is generally around 10%. At 7 mm away from the surface of the box girder, under the condition of 0.8ft + CH, the difference rate between point 1 and point 5 is between 15%–18%, and the difference rate between the remaining measurement points is less than 10%. When corroded only by chloride ion, the difference rate is about 10%.

Under the erosion condition of 0.8ft + CA + CH, when the original model does not consider the effect of carbonation, as shown in Figure 8a, the difference rate at 4 mm away from the surface of the box girder is generally more than 10%, and the maximum difference rate is 25.5%. After introducing the carbonation effect coefficient, the improved model in this paper can reduce the difference rate by 4% to 19%. Except for the difference rate of No. 9 measuring point reaching 14.2%, the difference rate of all other measuring points is less than 8%. At 7 mm away from the surface of the box girder, we can see from Figure 8b that the difference rate of the original model is between 4.5% and 29.8%. The improved model in this paper can reduce the difference rate to within 10%, and the maximum drop rate is 18%. The difference rate of the Cl^−^ model considering the effect of carbonation in this paper is generally within 10%, which is more accurate than the original calculation model.

## 4. Conclusions

This paper mainly carried out two parts of work. First of all, to quantitatively analyze the effect of carbonation on the diffusion of Cl^−^; the original formula for calculating the chloride ion concentration was modified by introducing the carbonation effect coefficient, and the calculation formula for the concentration of Cl^−^ considering the effect of carbonation is obtained. Secondly, by making a scale model of the concrete box girder, the durability deterioration test of the concrete box girder considering the combined effects of load, carbonation and chloride ion, was carried out, and the influence of carbonation and load on the diffusion of Cl^−^ in the concrete box girder was studied. The main conclusions are as follows:

The unevenness of chloride ion concentration in each part of the box girder is greater under load. The compressive stress can inhibit the diffusion of chloride ions; the tensile stress promotes the diffusion of Cl^−^ in the concrete. The maximum decrease (increase) of the free Cl^−^ content in the box beam under compression (tensile) stress is 17%.

Under the exposure conditions of 0.8ft + CA + CH and 0.8ft + CH, the unevenness of chloride ion concentration in each part of the box girder is generally close, and the free chloride ion content of the top and bottom of the test box girder extra eroded by carbonation increased from 23.5% to 33.5%. The maximum increase in the chloride ion concentration of the web was 26.7%. Carbonation can more significantly promote the diffusion of chloride ions.

The free chloride ion content of each measuring point of the box girder roof plate is consistent with the distribution law of the shear lag effect of the roof plate under the same load conditions. The free chloride ions of the box beam bottom plate are affected by the spatial structure characteristics of the box girder and the shear lag effect, and the content also showed an uneven distribution of “shear lag effect”.

The comparative analysis based on the test results shows that under the combined effect of load, CO_2_ and Cl^−^, the maximum difference rate of the original model is 29.8%. The improved model in this paper can reduce the difference rate by up to 18%, thus keeping the margin of difference within 10%.

## Figures and Tables

**Figure 1 materials-13-02117-f001:**
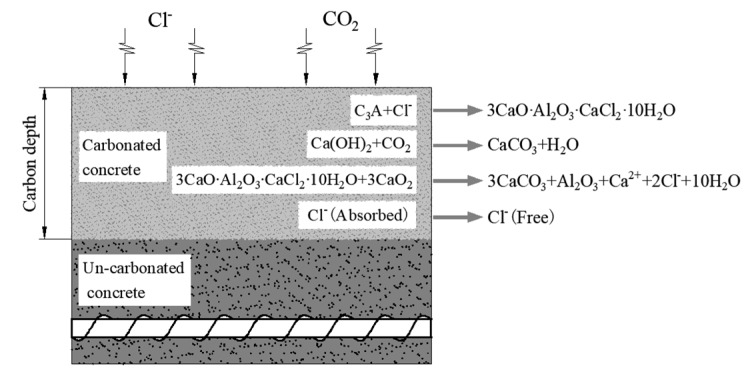
Interaction of chloride ion and carbon dioxide in concrete.

**Figure 2 materials-13-02117-f002:**
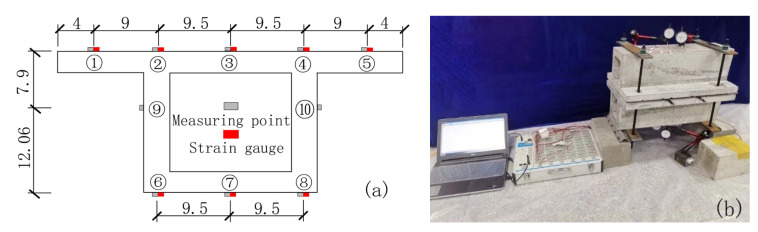
Model box girder measuring points and strain gauge arrangement (unit: cm). (**a**) Arrangement of measuring points and strain gauges; (**b**) Loading method.

**Figure 3 materials-13-02117-f003:**
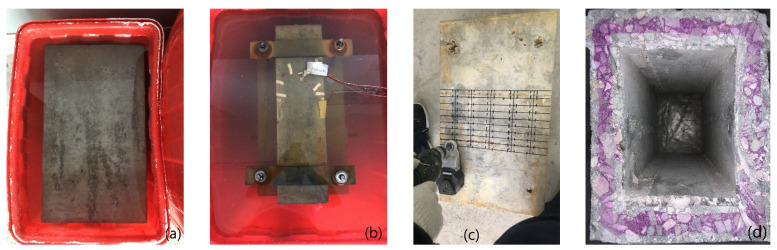
Corrosion process: (**a**) CH erosion; (**b**) ft + CH erosion; (**c**) Sample extraction; (**d**) Carbonation depth.

**Figure 4 materials-13-02117-f004:**
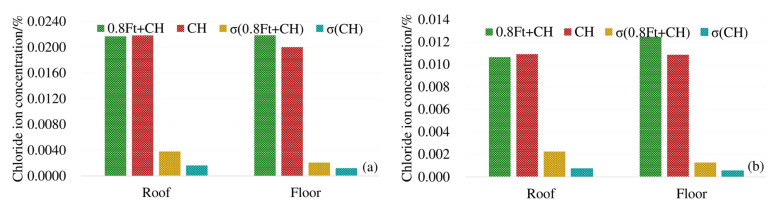
Average chloride concentration at different positions from the box girder surface: (**a**) 4 mm from the box girder surface; (**b**) 7 mm from the box girder surface.

**Figure 5 materials-13-02117-f005:**
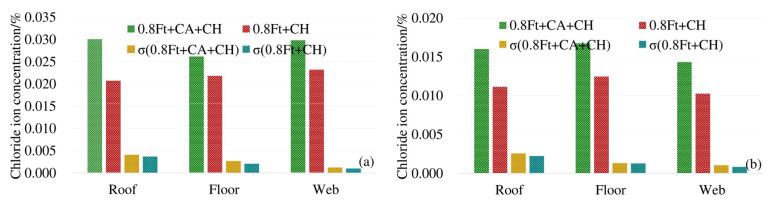
Average chloride concentration at different positions from the box girder surface: (**a**) 4 mm from the box girder surface; (**b**) 7 mm from the box girder surface.

**Figure 6 materials-13-02117-f006:**
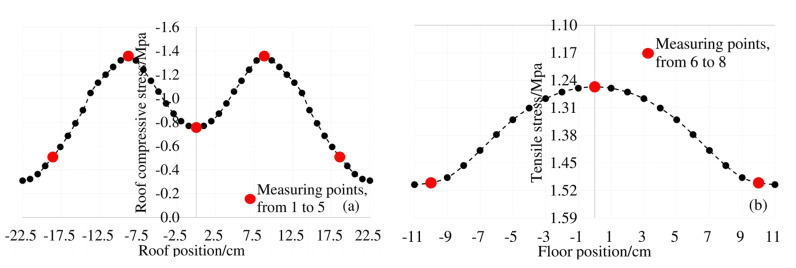
Shear lag effect of top and bottom plate under 0.8ft load: (**a**) stress distribution of roof; (**b**) stress distribution of floor.

**Figure 7 materials-13-02117-f007:**
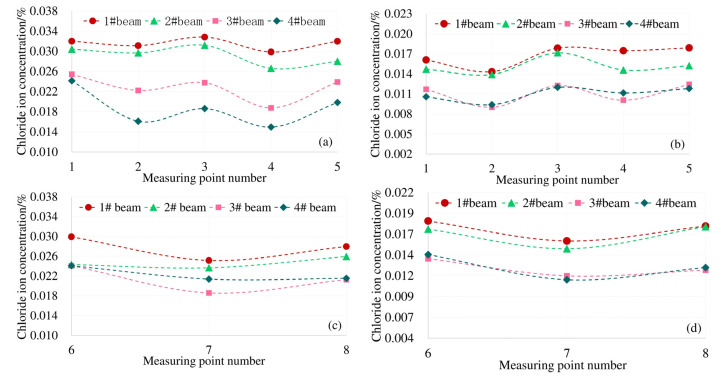
Chloride ion distribution on top and bottom plates affected by shear lag effect of box beam, (**a**) Concentration on the roof at 4 mm; (**b**) concentration on the roof at 7 mm; (**c**) concentration on the floor at 4 mm; (**d**) concentration on the floor at 7 mm.

**Figure 8 materials-13-02117-f008:**
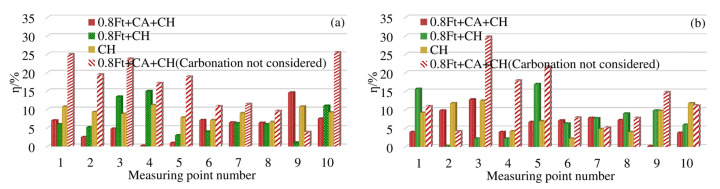
*η* of chloride concentration at each measuring point of the test box girder: (**a**) *η* at 4 mm; (**b**) *η* at 7 mm.

**Table 1 materials-13-02117-t001:** Chloride ion erosion test conditions and specimen grouping of box girder model.

No.	Working Type	Test Method	Test Beam Number
①	CH	Only the chloride ion corrosion test was performed: The unloaded box girder model was placed in a 5% sodium chloride solution for 45-Day and the test was stopped.	5#
②	0.8ft + CH	Chloride ion and load erosion test: The self-anchored loading method was used to load the box girder model, and the applied load was 0.8ft. The test was stopped after the box girder model in the loaded state was bubbled in a 5% sodium chloride solution for 45 days. It is the same as the exposed environment of test ①, except that the box beam model applies a bending load of 0.8ft.	3#, 4#
③	0.8ft + CA + CH	Chloride, carbonation and load erosion tests: The self-anchored loading method was used to load the box girder model, and the applied load was 0.8ft. The test box beam group under load is first placed in a carbonation box to accelerate carbonation for 48 h, and then taken out of the carbonation box and then placed in a 5% sodium chloride solution to infuse for 48 h, and then taken out and dried in an oven. After that, it is put in the carbonation box to accelerate the carbonation. The test was stopped alternately in such a way that the total carbonation time of the box girder model under the load reached 40-Day and the chloride ion erosion time reached 45-Day. The temperature, humidity, and carbon dioxide concentration of the carbonation test were 20 °C, 70%, and 20%, respectively. The chloride ion erosion was the same as the exposure environment and time of the test group ②.	1#, 2#

ft is the ultimate tensile strength of concrete; CA is the carbonation erosion; CH is the chloride ion erosion.

**Table 2 materials-13-02117-t002:** Performance index of 52.5 grade Portland cement.

Strength Grade	Fineness (%)	Cl^−^ (%)	SO_3_ (%)	MgO (%)	Setting Time (min)		Compressive Strength (MPa)		Flexural Strength (MPa)	
					Initial Setting	Final Coagulation	3-Day	28-Day	3-Day	28-Day
52.5	1.6	0.01	2.39	1.36	151	212	30.2	56.0	6.1	8.8

**Table 3 materials-13-02117-t003:** Concrete mix ratio.

Strength Grade	Water-Cement Ratio	Sand Rate (%)	Water Reducing Agent (%)	The Mixing Ratio (kg/m^3^)			
				Water	Cement	Sand	Gravel
C50	0.37	28	0.5	190	527	484	1245

**Table 4 materials-13-02117-t004:** Performance index of FDN superplasticizer.

Water Reduction Rate (%)	Bleeding Rate (%)	Gas Content (%)	Setting Time (min)	f (%)	s (%)	Na_2_SO_4_ Content	Solid Content
20	35	2.2	0.5	140	527	≤20	≥90

f is 28-Day compressive strength ratio; s is 28-day shrinkage ratio.

**Table 5 materials-13-02117-t005:** Stress and deformation during the loading process and the 28-day compressive strength.

Measuring Point Number	6	7	8	Mean Value
stress (MPa)	1.488	1.385	1.401	1.425
Vertical deformation (mm)	0.133	-	0.119	0.126
Test block number	1	2	3	mean value
28-Day compressive strength (MPa)	51.5	49.6	51.0	50.7

**Table 6 materials-13-02117-t006:** Measurement results of chloride ion inside the box beam under different erosion effects (%).

Structural Part	Point Number	Distance from Surface (mm)	0.8ft + CA + CH		0.8ft + CH		CH
			1# Beam	2# Beam	3# Beam	4# Beam	10# Beam
roof	1	4	0.03201	0.0257	0.02742	0.02414	0.02109
		7	0.01442	0.01429	0.01179	0.01067	0.00932
	2	4	0.03111	0.03042	0.02521	0.0161	0.02041
		7	0.01618	0.0145	0.0091	0.00949	0.00999
	3	4	0.03283	0.03218	0.02476	0.01861	0.02014
		7	0.01795	0.01725	0.01236	0.01209	0.01029
	4	4	0.02988	0.02659	0.0249	0.01494	0.02258
		7	0.01756	0.01467	0.01254	0.01191	0.00906
	5	4	0.03199	0.02801	0.01176	0.01983	0.0218
		7	0.01799	0.01531	0.01016	0.01125	0.01085
floor	6	4	0.02992	0.02432	0.02406	0.02408	0.02143
		7	0.01808	0.01474	0.01356	0.01404	0.01103
	7	4	0.02515	0.02365	0.02127	0.02137	0.01885
		7	0.01568	0.0171	0.01148	0.01102	0.01024
	8	4	0.02795	0.02592	0.01858	0.02153	0.01978
		7	0.01746	0.01739	0.01218	0.01249	0.01127
web	9	4	0.03201	0.0257	0.02742	0.02414	0.02109
		7	0.01242	0.01429	0.01179	0.01067	0.00932
	10	4	0.03111	0.03042	0.02521	0.0161	0.02041
		7	0.01618	0.0145	0.0091	0.00949	0.00999

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
