# Peer review of "Diffusion Behavior of Chloride Ions in Concrete Box Girder under the Influence of Load and Carbonation"

_materials, 2020, doi:10.3390/ma13092117_

Round 1

Reviewer 1 Report

The manuscript needs major revision as lot of things are not so clear in this current version. Please consider the following comments in revising the manuscript.

  • What the ultimate tensile strength Ft of concrete was as mentioned in table 1? And how the applied load 0.8Ft was decided, please discuss briefly.
  • Did you measure deformation or strain on the specimens due to the applied load? What about the cracking? Are there any data available on the crack properties?
  • Tell us more about the chloride ion test. How did you collect the samples? Is it by drilling or cutting? What was the drilling depth? What method was followed for determining the chloride ion? You can follow the following paper related to this work

https://doi.org/10.1016/j.conbuildmat.2016.03.206

  • I am also not so convinced about your statement “The results show that due to the effects of shear lag and the spatial structural characteristics of the box girder, the chloride ion concentration in the same section of the box girder differs greatly” in the abstract. I think the difference is not due to the shear lag and spatial. It could be due to cracking or surface strain level.
  • It’s also not easy to follow all the equations have shown in the manuscript. Did the authors develop any empirical formula from the results found here? You may be interested to follow the following papers in this regard. https://doi.org/10.1016/j.acme.2016.04.016 https://doi.org/10.1016/j.corsci.2012.11.028
  • In figures 4, 5 and 7, are these total and free chloride? Please mention it in the captions or in the axis title.
  • Please rewrite the conclusion section again by highlighting the important outcomes of this research.

Author Response

Dear Reviewer:
  Thank you for your letter and for the reviewers’ comments concerning our manuscript entitled Diffusion Behavior of Chloride Ions in Concrete Box Girder Under the Influence of Load and Carbonation. Those comments are all valuable and very helpful for revising and improving our manuscript, as well as the important guiding significance to our researches. We have studied comments carefully and have made correction which we hope meet with approval. The main corrections in the manuscript and the responds to the reviewer’s comments are as following:

comments:
1. What the ultimate tensile strength Ft of concrete was as mentioned in table 1? And how the applied load 0.8Ft was decided, please discuss briefly.

2.Did you measure deformation or strain on the specimens due to the applied load? What about the cracking? Are there any data available on the crack properties?

3.Tell us more about the chloride ion test. How did you collect the samples? Is it by drilling or cutting? What was the drilling depth? What method was followed for determining the chloride ion? You can follow the following manuscript related to this work

https://doi.org/10.1016/j.conbuildmat.2016.03.206

4.I am also not so convinced about your statement “The results show that due to the effects of shear lag and the spatial structural characteristics of the box girder, the chloride ion concentration in the same section of the box girder differs greatly” in the abstract. I think the difference is not due to the shear lag and spatial. It could be due to cracking or surface strain level.

5.It’s also not easy to follow all the equations have shown in the manuscript. Did the authors develop any empirical formula from the results found here? You may be interested to follow the following manuscript s in this regard. https://doi.org/10.1016/j.acme.2016.04.016 https://doi.org/10.1016/j.corsci.2012.11.028

6.In figures 4, 5 and 7, are these total and free chloride? Please mention it in the captions or in the axis title.

7.Please rewrite the conclusion section again by highlighting the important outcomes of this research.

Response to reviewer:
1. We are very sorry for our unclear report in Ft. According to China's national industry standard Code for Design of Concrete Structures, the Ft of C50 concrete is taken to be 1.89MPa. Supplementary explanations have been made on lines 215-216 of the manuscript.

2. A vertical deformation test was carried out during loading, but no cracks were detected. The specific deformation observation results have been reflected in Table 5 of the manuscript.

3. The content of free chloride ion is tested by chemical chloride testing. The specific test method can be referred to [36]. After the chloride ion erosion, a 12mm drill bit is used for borehole sampling, each sample is not less than 6 grams, and 20 samples are taken for each test point, and the average value is taken as the free chloride ion concentration of the test point. In this test, samples were taken at two depths of 4mm and 7mm from the surface. It has been added in lines 208-212 of the manuscript.

4. I also very much agree with you. Because the top plate is under compression, and under 0.8Ft load conditions, the compressive stress of the box girder top plate is small, the peak stress is -1.38MPa, and the possibility of cracks is small. In addition, under 0.8Ft load conditions, cracks may also appear on the surface of the bottom plate, but the distribution law of each measuring point 7mm away from the surface is also the same as the distribution law of the shear lag effect. The curve of the chloride ion content in the top plate in Fig. 7 is in good agreement with the distribution curve of the shear lag effect in Fig. 6 for the box girder. The tensile stress of the bottom plate is the largest under the action of shear lag, and there is two-dimensional diffusion at the measurement points No. 6 and No. 8.8. Under the combined action of the two, the chloride ion concentration distribution at each measuring point at 4 mm and 7 mm from the bottom plate surface is the same as the shear lag effect. Therefore, the conclusion that the reason for the difference in the chloride ion concentration of the same section of the box beam roof is caused by the shear lag effect and the space effect of the box beam is obtained. Chapter 3.1.3 of the manuscript has re-described the problem.

5. The revised manuscript has supplemented this content and compared with the original model to verify the improvement effect of the model in this article. The specific revised manuscript is shown in Chapter 3.2.

6. It is our negligence and we are sorry about this. The chloride ions in Figures 4, 5 and 7 are all free chloride ions, which are indicated in the title.

7. As the Reviewer's good advice, the revised manuscript has restated the conclusion based on the main research content of the manuscript .

Special thanks to you for your good comments.

Reviewer 2 Report

The authors report the study of concrete box girder degradation due to chloride-induced corrosion, load and/or carbonation. They prepared a scale model of concrete box girder and evaluated the chloride ion concentration in the structure, taking into account the effect of carbonation and shear lag. The authors refer to the effect of carbonization in the entire manuscript, including the title. I guess they actually mean carbonation. It seems quite a serious negligence. They prepared some specimens which were aged in different conditions before evaluating chloride presence in various spots. I am unsure how the chloride concentration was experimentally measured for the comparison with the model. The authors propose a new model for the calculation of chloride concentration in concrete box grinder.

Concerning the results, it seems important to discuss the experimental values taking into account the standard deviation associated to the measurements, to clearly state if the observed differences can be assigned to chloride/carbon dioxide/tensile strength. A comparison with previous models would also help to clarify the improvement here achieved.

In general, the work is interesting and the authors achieved some nice results, but the manuscript is not clearly written and it is hard to follow the discussion. The methods are not clearly presented, and it is difficult to gain a precise picture of the novelty of this work. Sometimes it is hard to understand what the authors wanted to say and there are many oversights. It seems that the manuscript was not accurately reviewed: chemical products are presented without taking care of subscripts; figures and captions present discrepancies; in many sentences, the dot at the end of the sentence is missing; the style of the manuscript is not coherent.

In addition to the major concerns detailed above, I report here below are some specific comments:

  • Section 3.2 is unclear to me. How did you decide the values here chosen for the coefficients in eq. 14? Is the difference λ representative to evaluate the accuracy of the model? Was it already used to validate new models?
  • In the abstract, the authors report that “the experimental value of the model proposed in this paper is generally within 10%, the accuracy is better”. Saying that the accuracy is “better” does not seems meaningful.
  • I would recommend the authors to reconsider the keywords they chose. “Experimental study”, e.g., indicates nothing in particular and does not help at all to describe this manuscript.
  • Lines 24-25: I am unsure what “economy and sustainable development” means in this contest.
  • Lines 27-29: the presence of chloride containing deicing salts and chloride induced erosion refer to the same process. This sentence is unclear and the reference is cited in a different style.
  • Line 29: the verb is missing in this sentence: “Once the corrosion of the reinforcement in the concrete, not only…”.
  • Lines 33-35: the authors claim that “the probability of damage to concrete members (?) due to insufficient durability increases by 16% due to climate changes, and the thickness of carbonization (?) increases by 45%”. Reported in this way this is confusing and incorrect. The cited papers refer to models and suggested that carbonation-induced damage risks may increase by more than 16% in Australia, and that for unloaded, non-pozzolanic concrete, ultimate carbonation depths in Toronto and Vancouver could be up to 45% higher.
  • Lines 37-40, 52-54: again, these sentences are confusing and I am unsure what the authors meant.
  • Lines 68-69: why don’t you report the main results present in the literature about the quantitative description of structural damage due to CO2 and Cl--? As far as I know the literature on this topic is extensive, at least when considering structures of different geometry.
  • According to the journal guidelines, all equations should be numbered. Here some equations are numbered, others not. Moreover, if you present some formulas, you should explain all the variables that are present, to avoid the readers an extensive search in the literature. As presented, this section is extremely hard to understand for those not experts in the specific field.
  • To perform the experiments here presented, did you follow some certificated quality test previously used in analogous system? Did you used recognised conditions of test?
  • Figure 1: is this figure taken from the literature? If yes, refer to the proper citation.
  • Figure 2: you refer to A) and b) in the caption. Beside the stylistic difference, in figure you did not indicated A/B.
  • Table 2: did you measure all the information reported in this table and for sand and gravel? If not, refer to the proper citation/manufacturer.
  • What about the water reducing agent? According to the journal guidelines, you should provide sufficient detail to allow others to replicate your experiments.
  • Figure 3: Again, you refer to A) b) c) d) in the caption. Again, you used different styles and you did not indicate a/b/c/d in figure.
  • Figures 4 and 5: Can you report your results together with error bars/standard deviation? Without an indication of the error associated to the experiments and heterogeneity, it is hard to assess if these differences are significant. Again, a) and b) are not indicated in figure.
  • Double check refuses. Carbonization is repeatedly used in the entire paper. Check superscript/subscript: most chemical formulas are inaccurate. Avoid repetitions. In few cases, dots at the end of the sentences are missing. At some point, the style of the manuscript changes…
  • Many references are not correctly cited. Check the reference style suggested in the journal instruction for authors. The numbers in the reference list are repeated twice.
  • Include a comparison with previous models to help clarifying the improvement here achieved.

Author Response

Dear Reviewers
  Thank you for your letter and for the reviewers’ comments concerning our manuscript entitled Diffusion Behavior of Chloride Ions in Concrete Box Girder Under the Influence of Load and Carbonation. Those comments are all valuable and very helpful for revising and improving our manuscript, as well as the important guiding significance to our researches. We have studied comments carefully and have made correction which we hope meet with approval. The main corrections in the manuscript and the responds to the reviewer’s comments are as following:

comments:

  1. The authors refer to the effect of carbonization in the entire manuscript, including the title. I guess they actually mean carbonation. It seems quite a serious negligence.
  2. I am unsure how the chloride concentration was experimentally measured for the comparison with the model.
  3. Concerning the results, it seems important to discuss the experimental values taking into account the standard deviation associated to the measurements, to clearly state if the observed differences can be assigned to chloride/carbon dioxide/tensile strength. A comparison with previous models would also help to clarify the improvement here achieved.
  4. The manuscript is not clearly written, and it is hard to follow the discussion. The methods are not clearly presented, and it is difficult to gain a precise picture of the novelty of this work. Sometimes it is hard to understand what the authors wanted to say and there are many oversights. It seems that the manuscript was not accurately reviewed.
  5. chemical products are presented without taking care of subscripts.
  6. figures and captions present discrepancies.
  7. in many sentences, the dot at the end of the sentence is missing; the style of the manuscript is not coherent.
  8. Section 3.2 is unclear to me. How did you decide the values here chosen for the coefficients in eq. 14? Is the difference λ representative to evaluate the accuracy of the model? Was it already used to validate new models?
  9. In the abstract, the authors report that “the experimental value of the model proposed in this manuscript is generally within 10%, the accuracy is better”. Saying that the accuracy is “better” does not seems meaningful.
  10. I would recommend the authors to reconsider the keywords they chose. “Experimental study”, e.g., indicates nothing in particular and does not help at all to describe this manuscript.
  11. Lines 24-25: I am unsure what “economy and sustainable development” means in this contest..
  12. Lines 27-29: the presence of chloride containing deicing salts and chloride induced erosion refer to the same process. This sentence is unclear and the reference is cited in a different style.
  13. Line 29: the verb is missing in this sentence: “Once the corrosion of the reinforcement in the concrete, not only…”..
  14. Lines 33-35: the authors claim that “the probability of damage to concrete members (?) due to insufficient durability increases by 16% due to climate changes, and the thickness of carbonization (?) increases by 45%”. Reported in this way this is confusing and incorrect. The cited manuscript s refer to models and suggested that carbonation-induced damage risks may increase by more than 16% in Australia, and that for unloaded, non-pozzolanic concrete, ultimate carbonation depths in Toronto and Vancouver could be up to 45% higher.

  15  Lines 37-40, 52-54: again, these sentences are confusing and I am unsure what the authors meant.

  1. Lines 68-69: why don’t you report the main results present in the literature about the quantitative description of structural damage due to CO2 and Cl--? As far as I know the literature on this topic is extensive, at least when considering structures of different geometry.
  2. According to the journal guidelines, all equations should be numbered. Here some equations are numbered, others not. Moreover, if you present some formulas, you should explain all the variables that are present, to avoid the readers an extensive search in the literature. As presented, this section is extremely hard to understand for those not experts in the specific field.
  3. To perform the experiments here presented, did you follow some certificated quality test previously used in analogous system? Did you used recognized conditions of test?
  4. Figure 1: is this figure taken from the literature? If yes, refer to the proper citation.
  5. Figure 2: you refer to A) and b) in the caption. Beside the stylistic difference, in figure you did not indicated A/B.
  6. Table 2: did you measure all the information reported in this table and for sand and gravel? If not, refer to the proper citation/manufacturer.
  7. What about the water reducing agent? According to the journal guidelines, you should provide sufficient detail to allow others to replicate your experiments.
  8. Figure 3: Again, you refer to A) b) c) d) in the caption. Again, you used different styles and you did not indicate a/b/c/d in figure.
  9. Figures 4 and 5: Can you report your results together with error bars/standard deviation? Without an indication of the error associated to the experiments and heterogeneity, it is hard to assess if these differences are significant. Again, a) and b) are not indicated in figure.
  10. Double check refuses. Carbonization is repeatedly used in the entire manuscript. Check superscript/subscript: most chemical formulas are inaccurate. Avoid repetitions. In few cases, dots at the end of the sentences are missing. At some point, the style of the manuscript changes…

  26  Many references are not correctly cited. Check the reference style suggested in the journal instruction for authors. The numbers in the reference list are repeated twice.

  27. Include a comparison with previous models to help clarifying the improvement here achieved.

 Response to reviewer:

1. We are very sorry for our negligence. The revised manuscript has been corrected.

  1. Chapter 3.2 of the revised manuscript is supplemented with relevant content. According to the test results, the difference between the original model and the improved model is compared to test the effect of the improved model.
  2. The specific deviations caused by carbonization and loading have been supplemented. Chapter 3.2 adds a comparison result of the analysis accuracy with the original model.
  3. Lines 208-228 of the revised manuscript supplement the loading and testing methods. Chapter 3.2 adds the comparative analysis content with the original model.
  4. The revised manuscript has been carefully checked and corrected.
  5. The revised manuscript has been carefully checked and corrected.
  6. The revised manuscript has been carefully checked and corrected.
  7. We are very sorry for this. This is a description error. In order to avoid misleading, the difference of formula (14) has been expressed as Æž. And Æž is used to verify the improved formula of chloride ion concentration in this manuscript.
  8. The revised manuscript has re-examined the sentence.
  9. The revised manuscript has reselected the keywords.
  10. The revised manuscript has deleted this statement.
  11. The revised manuscript has deleted this statement.
  12. The revised manuscript has re-examined the sentence.
  13. The revised manuscript has been re-expressed by quoting other literature.
  14. The revised manuscript has re-examined the sentence.
  15. The revised manuscript has been discussed again with reference to relevant literature.

  17.The formula has been carefully numbered and the meaning of the variables in the formula has been explained.

  18.The revised manuscript has supplemented the test process. The test method followed the previously recognized test conditions and quoted the corresponding literature.

  19.This figure is generally referred to the manuscript [32], which has been quoted on line 91.

  1. The Figure number has been added to the corresponding Figure.
  2. Lines 186-189 of the revised manuscript has been supplemented with relevant information.

  22.Table 4 is supplemented, and each performance index of the water reducing agent is described in detail.

  23.The specific Figure number has been added.

  1. Thank you very much for your comments. I have followed your comments, given the percentage of specific differences, and added the Figure numbers.
  2. The revised manuscript has been carefully checked and the wrong expressions have been corrected.
  3. The revised manuscript reapplied the reference and the format of the reference has been revised.
  4. The revised manuscript has supplemented this content and compared with the original model to verify the improvement effect of the model in this article. The specific revised manuscript is shown in Chapter 3.2.

 Special thanks to you for your good comments.

Reviewer 3 Report

Diffusion behavior of Chloride Ions in Concrete Box Girder Under the Influence of Load and Carbonization – Liu et al.

General Comments:

The article studied diffusion characteristics of chloride ions in a concrete girder box under the combined effect of loading and carbon dioxide. The chloride ion concentration calculation was revised by including the effect of carbonization. The manuscript may be considered for potential publication in its current form after minor changes. Specific comments are provided below.

Specific Comments:

  • A brief description of loading method presented in Figure 2(b) should be provided.
  • Although results presented in Figures 4 and 5 are quantitative values, it will be helpful for the readers to present the difference between 0.8Ft + CH and CH in relative terms expressed in %.
  • Line 298: Formula (13) has been referred here. The difference (lambda) was defined earlier in Equation (14).
  • In order to increase impact of the paper, the authors are encouraged to discuss diffusion of chloride ion (under carbonation & load) across various beam cross sections. The efficacy of the proposed method and its application on scaling up will be a relevant discussion item.

Author Response

Dear Reviewer:
  Thank you for your letter and for the reviewers’ comments concerning our manuscript entitled Diffusion Behavior of Chloride Ions in Concrete Box Girder Under the Influence of Load and Carbonation. Those comments are all valuable and very helpful for revising and improving our manuscript, as well as the important guiding significance to our researches. We have studied comments carefully and have made correction which we hope meet with approval. The main corrections in the manuscript and the responds to the reviewer’s comments are as following:

comments:

  1. A brief description of loading method presented in Figure 2(b) should be provided.
  2. Although results presented in Figures 4 and 5 are quantitative values, it will be helpful for the readers to present the difference between 0.8Ft + CH and CH in relative terms expressed in %..
  3. Line 298: Formula (13) has been referred here. The difference (lambda) was defined earlier in Equation (14).
  4. In order to increase impact of the manuscript, the authors are encouraged to discuss diffusion of chloride ion (under carbonation & load) across various beam cross sections. The efficacy of the proposed method and its application on scaling up will be a relevant discussion item.

 Response to reviewer:

  1. The revised manuscript supplements the loading method shown in Figure 2 (b) and gives reference to specific operating methods.
  2. Relevant descriptions have been added to chapters 3.1.1-3.1.2 of the revised manuscript. The specific error value is given.
  3. We are very sorry for this. This is a description error. In order to avoid misleading, the difference of formula (14) has been expressed as Æž. And Æž is used to verify the improved formula of chloride ion concentration in this manuscript.
  4. Thank you very much for your comments. Section 3.1.3 of the revised manuscript only shows the different distribution of chloride ions under load. The follow-up work will make more models specifically for the impact of carbonization for experimental research. The revised manuscript has supplemented this content and compared with the original model to verify the improvement effect of the model in this article. The specific revised manuscript is shown in Chapter 3.2.

Special thanks to you for your good comments.

Round 2

Reviewer 1 Report

In section 2.2.3, some error exists inspecting, prefix, suffix.

Line 188, unit of bulk density will kg/m3.

Author Response

Dear Reviewer:
  Thank you for your letter and for the reviewers’ comments concerning our manuscript entitled Diffusion Behavior of Chloride Ions in Concrete Box Girder Under the Influence of Load and Carbonation. Those comments are all valuable and very helpful for revising and improving our manuscript, as well as the important guiding significance to our researches. We have studied comments carefully and have made correction which we hope meet with approval. The main corrections in the manuscript and the responds to the reviewer’s comments are as following:

Reviewer comments:
  1. In section 2.2.3, some error exists inspecting, prefix, suffix.

2.Line 188, unit of bulk density will kg/m3. 

Response to reviewer :
  1. We are very sorry for our negligence. The revised draft has been carefully revised.

      2.The revised draft has been corrected.

      In addition, in order to more accurately describe the influence of various factors on the diffusion of chloride ions, Figures 4 and 5 increase the standard deviation of the chloride ion concentration in each part of the box girder under different exposure conditions.The revised version of other issues has also been corrected.

      Special thanks to you for your good comments.

Reviewer 2 Report

The authors largely improved the article. The manuscript is now clearer and has been improved with further details and with a useful comparison with the original model, but it seems that the manuscript was not entirely accurately reviewed. I have to point out that English language still needs to be improved.

I report here only few examples (not exhaustive, just as examples):

- the sentences at lines 15-17 are unclear, this part is grammatically incorrect: “..the chloride ion concentration in the same section of the box girder is differences.”.

- at lines 40-43 this sentence is unclear: ” For the environment in which concrete structures are located, they are often restricted by limiting the water-to-binder ratio, Concrete strength, protective layer thickness and chloride ion permeability coefficient, as well as construction and construction measures improve durability.”

I also have some more small suggestions:

- at lines 39, 53, 55, 59, 65, 71, 108, 212, 213, 229, 235, 255, 330, 336, 341, 344 you used again carbonization instead of carbonation. I reported here only some examples. I already underlined this issue in my previous revision and you answered that the revised manuscript were corrected. You corrected many points, but not all of them. The article needs to be checked again to avoid these oversights and all the grammatical errors.

- at line 35 “The cost of corrosion in China was approximately 310 billion USD,..”, perhaps you meant in the last year?

- line 41: you wrote “Concrete” instead of “concrete” in the middle of a sentence.

- As I already said in my previous review (point 17), according to the journal guidelines, all equations should be numbered. Here some equations are not numbered (see for example the equations at line 11, line 119). I reported here only some examples.

- As I already said in my previous review (point 6), the style of the expressions for chemical products needs to be checked (see for example line 94: CO2 not CO2; line 92; Table 2). I reported here only some examples.

- Figures 4 and 5: would it be possible to report the errors associated to the experimental measurements? Since you performed more tests, you could present the average results (which is the value shown in Figures 4 and 5 I guess) and the associated standard deviation (error bars). This information would allow assessing if the differences you observed are actually significant.

Author Response

Dear Reviewer:
  Thank you for your letter and for the reviewers’ comments concerning our manuscript entitled Diffusion Behavior of Chloride Ions in Concrete Box Girder Under the Influence of Load and Carbonation. Those comments are all valuable and very helpful for revising and improving our manuscript, as well as the important guiding significance to our researches. We have studied comments carefully and have made correction which we hope meet with approval. The main corrections in the manuscript and the responds to the reviewer’s comments are as following:

Reviewer comments:
  1. I have to point out that English language still needs to be improved.

  2. the sentences at lines 15-17 are unclear, this part is grammatically incorrect: “..the chloride ion concentration in the same section of the box girder is differences.

  1. at lines 40-43 this sentence is unclear: ” For the environment in which concrete structures are located, they are often restricted by limiting the water-to-binder ratio, Concrete strength, protective layer thickness and chloride ion permeability coefficient, as well as construction and construction measures improve durability.”
  2. at lines 39, 53, 55, 59, 65, 71, 108, 212, 213, 229, 235, 255, 330, 336, 341, 344 you used again carbonization instead of carbonation. I reported here only some examples. I already underlined this issue in my previous revision and you answered that the revised manuscript were corrected. You corrected many points, but not all of them. The article needs to be checked again to avoid these oversights and all the grammatical errors.
  3. at line 35 “The cost of corrosion in China was approximately 310 billion USD,..”, perhaps you meant in the last year?
  4. line 41: you wrote “Concrete” instead of “concrete” in the middle of a sentence.
  5. As I already said in my previous review (point 17), according to the journal guidelines, all equations should be numbered. Here some equations are not numbered (see for example the equations at line 11, line 119). I reported here only some examples.
  6.  As I already said in my previous review (point 6), the style of the expressions for chemical products needs to be checked (see for example line 94: CO2 not CO2; line 92; Table 2). I reported here only some examples.
  7. Figures 4 and 5: would it be possible to report the errors associated to the experimental measurements? Since you performed more tests, you could present the average results (which is the value shown in Figures 4 and 5 I guess) and the associated standard deviation (error bars). This information would allow assessing if the differences you observed are actually significant.

 Response to reviewer:
  1. We are very sorry for our unclear report. The revised draft has been carefully revised.

  1. The revised draft has been restated.
  2. It has been restated on lines 55-57 of the revised draft.

    4 We are very sorry for our negligence. The revised manuscript has been corrected.

  1. The revised draft has been supplemented with complete information 6. It is our negligence and we are sorry about this. The chloride ions in Figures 4, 5 and 7 are all free chloride ions, which are indicated in the title.
  2. The revised draft has been restated.
  3. We are very sorry for our unclear report. We have carefully checked the manuscript and found that the place you indicated is not a formula. It is an explanation of the error function ( )of the corresponding formula, so it is not numbered. 
  4. The revised draft has been carefully checked and revised.
  5. Thank you very much for your suggestions. The revised draft supplements the standard deviation information in Figure 4 and Figure 5, and conducts relevant analysis based on this.

  Special thanks to you for your good comments.